# The Dynamic and Crucial Role of the Arginine Methylproteome in Myoblast Cell Differentiation

**DOI:** 10.3390/ijms24032124

**Published:** 2023-01-20

**Authors:** Nikolaos A. Papanikolaou, Marios Nikolaidis, Grigorios D. Amoutzias, Ariadni Fouza, Maria Papaioannou, Akhilesh Pandey, Athanasios G. Papavassiliou

**Affiliations:** 1Laboratory of Biological Chemistry, School of Medicine, Aristotle University of Thessaloniki, 54124 Thessaloniki, Macedonia, Greece; 2Bioinformatics Laboratory, Department of Biochemistry and Biotechnology, University of Thessaly, 41500 Larisa, Greece; 3Fifth Surgical Department, Ippokrateio General Hospital, School of Medicine, Aristotle University of Thessaloniki, 54643 Thessaloniki, Macedonia, Greece; 4Department of Laboratory Medicine and Pathology, Center for Individualized Medicine, Mayo Clinic, Rochester, MN 55905, USA; 5Manipal Academy of Higher Education (MAHE), Manipal 576104, Karnataka, India; 6Department of Biological Chemistry, Medical School, National and Kapodistrian University of Athens, 11527 Athens, Greece

**Keywords:** arginine, C2C12, differentiation, methylproteome, myoblast, SILAC, Short Linear Motifs (SliMs)

## Abstract

Protein arginine methylation is an extensive and functionally significant post-translational modification. However, little is known about its role in differentiation at the systems level. Using stable isotope labeling by amino acids in cell culture (SILAC) proteomics of whole proteome analysis in proliferating or five-day differentiated mouse C2C12 myoblasts, followed by high-resolution mass spectrometry, biochemical assays, and specific immunoprecipitation of mono- or dimethylated arginine peptides, we identified several protein families that were differentially methylated on arginine. Our study is the first to reveal global changes in the arginine mono- or dimethylation of proteins in proliferating myoblasts and differentiated myocytes and to identify enriched protein domains and novel short linear motifs (SLiMs). Our data may be crucial for dissecting the links between differentiation and cancer growth.

## 1. Introduction

Post-translational modifications (PTMs) such as acetylation, phosphorylation, and methylation are critical for the epigenetic regulation of gene expression and signal transduction in health and disease [1,2,3,4,5]. PTMs are involved in all signaling cascades and fine-tune cellular and systemic responses to internal and external cues [6,7,8]. The post-translational modification of histone or non-histone proteins and, in particular, the addition or subtraction of methyl groups from arginine is important in epigenetic inheritance. Enzymes that catalyze arginine methylation are drug targets whose functional modifications are currently under intense scrutiny in the treatment of diseases such as cancer [9,10,11,12]. The methylation of arginine side chains in non-histone proteins is emerging as a significant post-translational modification in the regulation of transcription [10,13,14,15], RNA splicing [16,17], DNA damage response [18,19,20,21], cellular differentiation [22,23,24,25], and cell death [26,27,28]. The transfer of one or more methyl groups from S-adenosylmethionine (AdoMet or SAM) to the guanidinium nitrogen atoms of arginine residues is catalyzed by protein arginine methyltransferases (PRMTs). The guanidine group of arginine contains five potential hydrogen bond donors that participate in interactions with biological hydrogen bond acceptors [29]. Studies on the HIV-1 Tat protein which specifically binds to a bulged region of TAR RNA, have shown that the arginine eta nitrogens and the epsilon nitrogen can hydrogen bond with adjacent pairs of phosphates and that these interactions occur near RNA loops but not within double-stranded A-form RNA. Thus, arginine side chains are employed for the recognition of specific RNA structures [30]. Protein arginine methylation is a critical modification with a multitude of effects on cell biology [14,21,31]. Mammals have a family of nine sequence-related protein arginine methyltransferases (PRMTs, PRMT1–PRMT9) that catalyze the transfer of a methyl group to the nitrogens in arginine. These enzymes modify a variety of proteins using the cofactor S-adenosylmethionine (AdoMet) to produce ω-NG monomethyl-arginine (MMA), ω-NG, NG asymmetric dimethyl-arginine (aDMA), ω-NG, and N7 symmetric dimethyl-arginine (sDMA). Protein arginine methylation regulates many cellular pathways, such as transcription, translation, and DNA damage repair [31].

The modification of arginine residues in proteins alters protein-binding interactions and can therefore regulate their biological functions. In DNA–protein interactions, arginine residues are the most frequent hydrogen bond donors to the four bases and phosphate groups [32]. The methylation of arginine can potentially alter the conformation of the protein targets, and because it neutralizes the potential hydrogen bond donors in proteins, it can also inhibit protein–protein interactions [11,33,34]. For example, the arginine methylation of Sam68 proline-rich motifs can inhibit its binding to SH3, but not to WW domains [35]. The methylation of arginine residues may also increase their affinity for aromatic rings in cation–pi interactions [36]. Such interactions are observed in the aromatic cage of the SMN Tudor domain, which likely interacts with the methylated tail of the SmD splicing factor [37]. The activation or repression of the proliferation and differentiation pathways is partly controlled by the post-translational modifications of proteins, which are instrumental in inducing these dramatic morphological changes [38,39]. For example, YAP, a main component of the Hippo pathway, is found in both myoblasts and myocytes. In proliferating myoblasts, YAP is localized in the nucleus and is hypophosphorylated at Ser127. Upon differentiation, there is a 20-fold increase in YAP Ser127 phosphorylation accompanied by translocation from the nucleus to the cytosol [40].

Skeletal muscle is generated from the multi-stage differentiation of the myoblast precursors of somite mesenchymal origin in late embryogenesis and comprises nearly 40% of total body mass in humans. Myoblasts are skeletal mononuclear myocytes that are committed to the myogenic lineage but are also capable of proliferation in cell culture [39]. During differentiation, myoblasts elongate and fuse with each other, forming multinucleated, terminally differentiated myocytes, capable of contraction. The induction of myocyte differentiation is initiated by the successive activation of signaling pathways that cooperate in inhibiting proliferation and in inducing differentiation via the expression of several muscle differentiation-specific transcription factors, such as the basic-helix-loop muscle regulatory Myf5 protein, which is expressed first in the developing embryo, followed by MyoD, myogenin, and MRF5 [41,42]. Cell cycle arrest is induced by activated retinoblastoma protein, as well as the cdk inhibitors WAF1/p21 and p27 and myogenin, and is followed by the differentiation-specific expression of markers such as troponin T [43,44,45,46].

Several pathways regulate the differentiation of C2C12 mouse myoblast cells. For example, the Notch pathway inhibits terminal differentiation through activation of the expression of the transcriptional regulator human C promoter Binding Factor (CBF1) [47]. In contrast, the histone acetyltransferase PCAF enhances the MyoD-mediated transcription and myogenic differentiation, presumably through complex formation with MyoD, leading to activation of the expression of WAF1/p21 [48]. ERK6, a member of the MAPK family of mitogen-activated serine/threonine protein kinases, and p38 activator MAP kinase (MKK3), have been implicated in C2C12 cell myogenic differentiation [49]. P38 kinase is responsible for the phosphorylation of myocyte enhancer factor 2 (MEF2A) and the subsequent activation of transcription leading to C2C12 differentiation [50]. Recently, the methylation of specific proteins has been implicated in the regulation of the cellular differentiation of mouse C2C12 myoblasts to myocytes. Thus, the lysine methyltransferase set7/9 is required for differentiation via direct interactions with MyoD [51]. The arginine methyl transferase 5 (PRMT5) is required for MyoD-induced myotube formation through interactions of the Brg1-associated ATPase with the SWI-SNF chromatin remodeling complex [52]. In contrast, the lysine methyltransferase G9a inhibits differentiation, at least partly through the methylation of the transcription factor MyoD [53,54,55]. In spite of these reports, we lack a comprehensive understanding of the role of arginine methylation on myoblast proliferation or in the transition to the differentiated state at a systems level. The differentiation of mouse C2C12 myoblasts is an ideal model to dissect the methylation of arginines on a proteomic scale and to identify the protein networks that regulate muscle cell differentiation.

In this study, we identified several different protein families that are differentially methylated in arginine and characterized proliferating or differentiated C2C12 cells. In addition, we identified the activated pathways to which these protein families belong, these proteinsbelong, the most prominent being the mRNA processing pathways. We identified key domains in these proteins and also SLiMs. Moreover, we found that the levels of key CDKs are reduced in the differentiated cells. These results identify several proteins whose arginine methylation status changes drastically in differentiated cells, and they pave the way towards the identification of methylation networks that characterize cell differentiation.

## 2. Results

### 2.1. Quantitative Changes in the Cellular Proteome during Myoblast Differentiation

We identified, for the first time, changes in the methylation of arginine on proteins in proliferating or differentiatingmouseC2C12 myoblast cells, according to the strategy shown in Figure 1. When switched from 20% to 2% FBS, mouse C2C12 myoblast cells differentiate within five days into elongated multinuclear myocytes, which are reminiscent of muscle cells. We used SILAC proteomics to probe the whole proteome and identify and quantify the protein families and proteins whose levels changed significantly during differentiation [56,57]. The proliferating cells were grown for at least five generations to ensure complete incorporation in the presence of “heavy” Lysine (^13^C6^15^N2) and Arginine (^13^C6^15^N4),and the differentiated cells were grown in “light” lysine (^2^H4) and arginine (^13^C6), respectively (Figure 1 and Materials and Methods). We determined global changes in the levels of proteins by whole proteome analysis on an LTQ-Orbitrap Elite mass spectrometer and by determining the heavy-to-light (H/L) ratio of the proteins, as described in Materials and Methods and according to the strategy in Figure 1. Heavy, non-radioactive Lysine 8 (K8) and arginine 10 (R10) were used in proliferating the myoblasts, whereas light K and R were used to maintain the differentiated myocytes for five days. The proteins were isolated and analyzed by LC-MS/MS, as described in the Materials and Methods section. Fragmentation spectra from three independent experiments were obtained for 4600 proteins, of which 4133 were accurately quantified by their H/L ratio (Appendix A). A high H/L ratio was indicative of downregulated proteins, whereas a low ratio was indicative of upregulated proteins [56,58]. Because heavy amino acids were used in proliferating the myoblasts only, an (H/L) ratio greater than 2 indicated a significant reduction in protein levels in the differentiated cells, whereas an H/L ratio smaller than 1 indicated increased protein levels in the differentiated cells.

### 2.2. Arginine Methylation Is Important for Myoblast Differentiation

We and others have previously established [59,60] that adenosine-2′,3′-dialdexyde (AdOX), an adenosine analog and a global S-adenosylmethionine-dependent methyltransferase inhibitor with an IC50 of 40 nM, inhibits C2C12 cell differentiation in the range of 5 to 50 μM. AdOX inhibits transmethylation through the accumulation of S-adenosylhomocysteine (SAH), a negative feedback inhibitor of methylation, through the suppression of SAH hydrolase (SAHH). The increase in SAH levels results in the feedback inhibition of most methylation reactions [33]. The proliferating C2C12 cells grown in heavy K8 and R10 amino acids appeared to display a typical mesenchymal myoblast cell morphology (Figure 2A, days 1 and 2, upper panels). As shown in the control, in the day 1 upper panel in Figure 2A, the differentiated cells displayed the typical elongated morphology of myocytes (Figure 2A, day 3 and day 5,upper panels). By the fifth day in the differentiation medium, almost all the cells had differentiated into myocyte-like cells.

To assess the role of protein methylation in myoblast differentiation, we treated the C2C12 cells grown in culture to confluency, which had begun the process of differentiation, with AdOX. No visible effects were observed on the cells at the low end of the concentration range [60]; however, the cells treated with 50 μM AdOX visibly exhibited reduced numbers of differentiated myoblasts compared to the controls (Figure 2A, day 3 and day 5, bottom panels). Notably, the differentiated cells in the 50 μΜ group were markedly smaller than those in the control group, indicating that the process was interrupted at a specific stage of the cell cycle. Our AdoX inhibition data suggest that methylation is required for the completion of the differentiation of mouse myoblasts.

### 2.3. Alterations in the Arginine Methylproteome Revealed by Quantitative Proteomics

We assessed the genome-wide changes in the status of arginine methylation by subjecting whole cell extracts (WCE) isolated from proliferating or five-day differentiating C2C12 cells to morphological (Figure 2A, control panels) and immunoblot analyses (Figure 2B) with highly specific antibodies that detect mono-methyl, symmetric, or asymmetric di-methyl arginine (henceforth referred to as MMA, SDMA, and ADMA; SDMA and ADMA will collectively be referred to henceforth as DiMA). Significant differences were observed in the overall protein arginine methylation status for several protein bands with anti-MMA, SDMA, and ADMA antibodies in one-dimensional polyacrylamide gels, ranging from 15 kDa, where histones migrate, to 140 kDa (Figure 2B, blue arrows). Specifically, a significant reduction in the overall methylation of arginine was observed in differentiated cells for all three methylation types (Figure 2B, three left panels; see the blue arrows pointing to reduced bands). As differentiation requires the interruption of the cell cycle, we evaluated the levels of cell cycle regulators that had high H/L ratios by immunoblotting WCE for CDK2 and CDK6, two key cell cycle regulating kinases with H/L ratios equal to 1.55 and 3.08, respectively (Figure 2B, right panels). Both the CDK2 and the CDK6 levels were substantially reduced in the differentiated C2C12 cells (Figure 3B, right panels, lane 2). In contrast, the levels of ERK1, a MAPK kinase that is also involved in the cell cycle, with an H/L ratio of 1.03, remained virtually unchanged (Figure 2B, right panels). We analyzed the data from three independent experiments using ANOVA to establish the significance of our findings by constructing volcano scatter plots and by setting a p-value equal to or smaller than 0.01 (Figure 3A). From this analysis, several proteins, whose levels changed dramatically between proliferating and differentiated cells, were identified.

### 2.4. Identification of Differentially Methylated Proteins on Arginine

We systematically explored differences in the levels and types of protein arginine methylation at the proteomics level by utilizing the unbiased, stable isotope-labeled (SILAC)-based quantitative methylproteomic approach [56], which employed the mixing of labeled wholecell extracts from proliferating and five-day differentiated C2C12 myoblasts (see Materials and Methods for details). As with the isolation of the total proteome, the proliferating cells were grown for at least five generations to ensure complete incorporation in the presence of “heavy” Lysine (^13^C6^15^N2) and Arginine (^13^C6^15^N4), and the differentiated cells were grown in “light” lysine (^2^H4) and arginine (^13^C6), respectively (Figure 1 and Section 4. Following cell lysis, purification, and digestion with trypsin (which cleaves next to lysine and arginine), equal amounts of the cell extracts from the proliferating and five-day differentiated cells were mixed and subjected to enrichment by the immunoprecipitation of symmetric or asymmetric mono- or di-methylated arginine peptides, using resin-conjugated monoclonal antibodies that are specific for MMA, ADMA, or SDMA, as described in the Materials and Methods section. To increase the reliability of our analyses, we performed the experiments in triplicate. The peptides were isolated and dried and subjected to LC-MS/MS analysis. High-quality tandem mass spectrometry data were acquired with high accuracy and resolution (resolution = 120,000 at 400 *m*/*z*) on an LTQ-Orbitrap Elite mass spectrometer. The data were then subjected to Mascot analysis at a false discovery rate of 1%. Substantial coverage was obtained for the proteome, with the precise quantitative measurements of over 4000 proteins. Mono-methyl KR antibodies detected quantitative methylation changes in 271 proteins between the proliferating and differentiated cells that were common in three independent experiments, whereas the symmetric di-methyl or asymmetric di-methyl antibodies detected changes in 23 and 62 proteins that were common in three independent experiments (Figure 3B).

### 2.5. Transcription Factors and Histones Enriched in MMA and in DiMA Samples

In both the mono-methylated (MMA) and di-methylated (DiMA) samples (SDMA and ADMA were identified as such with the specific antibodies only, as described in Section 4 and Appendix A), several regulatory proteins underwent differential methylation on arginine. Several transcription factors and histone types underwent MMA or DiMA. In the MMA samples of the differentiated cells, TFIID (H/L > 22), transcription elongation regulator isoform X5 (H/L > 14), general transcription factor 3C polypeptide 2, isoform 4 (H/L > 10), and p66 isoform b (H/L = 2.7) were heavily hypomethylated. H3.2 type 2-b histones are heavily hypomethylated in differentiated myocytes (H/L > 18). Our data showed that these histones were more methylated in the proliferating myoblasts. Interestingly, the arginine methylation of the TATA box-binding protein (TBP) was reduced in the differentiated cells (H/L ratio equal to 15). Several transcription and mRNA processing factors were also mono-methylated/demethylated in the proliferating and differentiating cells (Appendix A). Noteworthy among the other proteins was myosin-10, a non-muscle-specific myosin, which appears to be relatively MMA-enriched in differentiated cells (H/L = 0.6). In contrast, four transcription factor proteins were hypomethylated in proliferating myoblasts and comparatively overmethylated in differentiated myocytes, SPT5 (H/L < 0.4), transcription intermediary factor 1, isoform 2 (H/L < 0.04), BTF3, isoform 2 (H/L < 0.2), and Purine-Rich Element-Binding Protein B (Pur beta (H/L < 0.5)). In the DiMA samples, the transcription elongation factor SPT5 (H/L > 10) and TATA box-binding protein-associated factor RNA polymerase I subunit C isoform X2 (H/L > 15) were heavily hypomethylated in the differentiated myocytes. In contrast to the MMA samples in the differentiated cells, where it was substantially under-methylated, histone H3.2 was slightly over-dimethylated in arginine (H/L < 0.6) in the differentiated cells. It was also notable that two ATP-dependent RNA helicases were significantly under-methylated in the differentiated myocytes (H/L > 15). Interestingly, the proteins in the RNA processing pathways appear to be the predominant family of proteins that are significantly differentially methylated in arginine in the MMA or DiMA of differentiated cells (Figure 3A; also see Section 2.7 and Section 2.8). Notable examples include different isoforms of the Synpo 2l protein (synaptopodin 2-like protein isoform X3) (H/L = 0.03 in whole proteome and H/L ≤ 0.01–0.04 in MMA, SDMA, or ADMA samples), indicating that both its levels and its arginine methylation increased more than 33 times in the differentiated cells (whole proteome), as well as in the MMA, ADMA, or SDMA samples of the differentiated cells (Figure 3A and Appendix A). Two other proteins of notable significance are the Scrib and several Fubp proteins (see Section 3). Scrib is heavily dimethylated or monomethylated in differentiated cells (H/L = 0.009 and 0.047, respectively, Appendix A). In contrast, the Fubp isoforms are hypomethylated in differentiated cells (H/L between 25 and 18, Appendix A).

### 2.6. Pathways Enriched in Down- or Upregulated Proteins in Whole Proteome with SILAC Proteomics

We subjected upregulated or downregulated, whole proteome proteins (Appendix A) to gene ontology analysis using the Metascape program, which combines functional enrichment, interactome analysis, gene annotation, and membership search [61]. Using proteins with an H/L ratio equal to or lower than 0.20 for the upregulated proteins (levels increased in the differentiated cells 5 times, 122 proteins in total) and anH/L equal to or higher than 2.5 for the downregulated proteins in the differentiated cells (levels reduced in the differentiated cells, 260 proteins in total), we obtained the top-enriched pathway classes shown in Figure 4.

Not surprisingly, in the differentiated cells the cell cycle processes, ribosome biogenesis, and DNA replication are downregulated, whereas in the proliferating cells the respiration (the citric acid cycle and oxidative phosphorylation), lipid metabolism, mitochondrial biogenesis, and muscle growth proteins, among others, are increased.The levels of several CDKs, such as CDK1, CDK2, CDK6, CDK18, CDK13, and CDK11b (H/L ratios between 8 and 1.6) are drastically reduced (Appendix A). The immunoblots for CDK2 and CDK6 validated our SILAC results (Figure 2B, right panel).

### 2.7. Pathways Down-or Upregulated for Arginine Monomethylation (MMA) in Differentiated Cells

To provide a functional context on the pathways for the mono- or dimethylated proteins that are enriched in differentiated C2C12 cells, we undertook a more systematic analysis of the methyl-peptides recognized by the MMA or DiMA-arginine-specific antibodies by performing gene ontology and pathway analysis on the downregulated MMA and DiMA proteins in arginine methylation in the differentiated cells.First, we used the top 20 proteins that were mono- or di-methylated as queries in STRING and extracted their networks using 10 first-shell and 20 s-shell genetic and physical interactors with an E value equal to 0.7 or 0.9. We then used the lists of methylated protein interactors as queries in the annotation tool Metascape, which computes enrichment by also considering homologs and the networks of the protein lists in the query. This analysis revealed that the MMA pathways with arginine hypomethylated proteins (Appendix A) were enriched in RNA processing, pluripotency, mitotic cell cycle regulation, activated kinase, and other pathways. The top MMA pathways with enriched, upregulated arginine methylation include SRP-dependent co-translational proteins targeting membrane proteins, the 43S complex, and the ribosome assembly proteins. Interestingly, the hypomethylated (downregulated for arginine dimethylation) DiMA pathways were enriched for RNA metabolism, mRNA processing, and translation proteins. In contrast, the pathways upregulated for DiMAwere enriched for oxidative metabolism and for mRNA metabolism. 

### 2.8. Pathways Down- or Upregulated for Arginine Demethylation (DiMA) in Differentiated Cells

We also subjected to gene ontology and network analysis the groups of down- or upregulated proteins in the DiMA samples. The enriched proteins downregulated for arginine dimethylation belonged to RNA metabolism, translation, mRNA processing and, interestingly, parvulin-associated pre-rRNP complex. Interestingly, the proteins upregulated for arginine dimethylation also belonged to the mRNA processing pathways. In addition, the oxidative metabolism contained proteins that appeared to be enriched in dimethylated arginine.

### 2.9. SLiMS and Protein Domains Enriched in MMA or DiMA Proteins

We analyzed the top proteins in Section 2.6 and Section 2.7 above for enriched SLiMs or protein domains andidentified several novel SLiMs. The SLiMs and protein domains that are enriched in MMA or DiMA proteins are described in the Materials and Methods. The data are shown in Appendix A. The novel SLiMs for each category of sample (in different colors on the upper right side) were dispersed in various positions within the protein sequences (Figure 5 and Figure 6; see Appendix A for details on top SLiMs and their precise locations within the protein domain sequence).

We also extracted the protein domains that were enriched in these proteins (Appendix A and Table 1). The RRM_1 domain, which is found in RNA-binding proteins, was the most abundant domain (28 occurrences in 12 out of 49 proteins, 60%) in all four categories of samples. The second most abundant domain is the KH domain, which was first discovered in the heterogeneous nuclear ribonucleoprotein(hnRNP). It occurs at a frequency of 20% (12 occurrences in 12 out of 49 proteins). The third most abundant domain is the PDZ domain, which is found 9 times in 2 out of 49 proteins (18%) and is present in the proteins anchored to cell membrane organizing signaling complexes (Appendix A, Table 1).

## 3. Discussion

The arginine methylation of histones is an epigenetic mechanism that regulates gene expression in eukaryotic cells. The methylation of arginines in non-histone proteins is also part of these mechanisms; however, there is a dearth of information on a systems level. The addition of methyl groups to the arginine residues of histone and non-histone proteins is catalyzed by a family of nine protein arginine methyltransferases, resulting in either mono- or dimethylatedarginine residues [3,62,63,64]. Similarto other post-translational modifications, methylation also appears to be a reversible modification. However, little is known about the enzymes that catalyze arginine demethylation. Moreover, the role of enzymes in catalyzing the direct removal of methyl groups from arginine residues in proteins has been controversial. At least two different families of enzymes have been recognized as putative demethylases: peptidyl arginine deiminases andJumonji domain-containing proteins [64,65,66,67].

In this study, we sought to identify, for the first time, the proteins whose levels are significantly differentially regulated in differentiated C2C12 mouse myocyte cells and to specifically identify the proteins that are modified by arginine methylation on a proteomics scale.We performed an immunoprecipitation analysis of the methylated peptides derived from C2C12 cells that proliferated or differentiated for five days, followed by high-resolution MS/MS. This approach has been particularly successful in analyzing post-translational modifications of proteins in different physiological contexts [68]. Mouse C2C12 myoblast cells undergo morphologically well-defined differentiation into elongated myocytes that can be triggered by serum withdrawal (see Section 4). The well-defined and distinct morphological changes that reflect the distinct changes in the expression programs from myoblasts to myocytes, as well as their reproducibility, render C2C12 cells ideal for this type of analysis of differentiation using proteomics methods [56].

The biological role of non-histone arginine methylation during the differentiation of myoblasts into myocytes is poorly understood. MEF2A is the only known differentiation-specific factor whose activity is regulated by the methylation/demethylation of a specific lysine residue during differentiation [69]. The cultured C2C12 cells were morphologically analyzed for changes in shape and size. The proliferating C2C12 cells (maintained at a density of no more than 40–70%) appeared to display a typical mesenchymal myoblast cell morphology (Figure 2A, day 1, upper left panel), whereas the differentiated cells displayed the typical elongated morphology of myocytes (Figure 2A, days 3 and 5). By the fifth day in the differentiation medium, almost all the cells had differentiated into multinuclear myocytes. Significant differences were detected in the methylation status of several protein bands in the one-dimensional polyacrylamide gels in the range of 15 kDa, where histones migrated, and up to 140 kDa (Figure 3B, left panels) for all the antibodies used. Specifically, a reduction in overall methylation was detected in the differentiated cells. This finding is consistent with the generally proposed role of arginine methylation in cell differentiation [43,46,54,70,71]. Because the proteins were quantitated solely by the H/L ratio in the whole proteome (and separately in independent experiments with anti-MMA, anti-ADMA, or anti-SDMA monoclonal antibody immunoprecipitation), ADMA and SDMA were indistinguishable and collectively referred to as DiMA, as discussed in the next section.

The analysis of the MMA and DiMA samples revealed that the most frequent proteins undergoing differential mono- or dimethylation overwhelmingly belonged to the RNA processing pathways that involve interactions with RNAs (Figure 4A,B and Table 1). The most frequent domain was the RRM_1 domain, which was enriched in all four categories of the arginine-methylated peptides identified (Table 1). The RNA recognition motif (RRM) is an RNA-binding domain of about 90 amino acids that binds single-stranded RNAs and has been implicated in pathological conditions [72,73,74,75]. RRM domains are found in many RNA-binding proteins, such as heterogeneous nuclear ribonucleoproteins (snRNPs) and splicing proteins [76,77]. Our data are consistent with early reports on the role of arginine methylation in RNA-binding proteins. The second most frequent domain found in arginine-methylated proteins is the PH domain, which is also found in RNA-binding proteins. RNA-binding proteins (RBPs) often contain glycine-arginine-rich (GAR) motifs and are considered major substrates for PRMTs. The methylation of GAR motifs influences the RNA–protein interactions by prohibiting the formation of hydrogen bonds by steric hindrance and causing the arginine residues to become more hydrophobic [78].

Several proteins that undergo differential arginine methylation were found. Fubp3 and Scrib are of particular interest because they are members of cell growth and proliferation networks. Although not fully established, Fubp3 appears to possess single-stranded DNA binding activity, is involved in the regulation of gene expression, and has recently been implicated in different cancers [79,80,81]. Moreover, Fubp3 interacts with 3′ microsatellite sequences and regulates translation [82]. The participation of Fubp3 in the Myc network, which must be inhibited for cell differentiation to occur, suggests an essential role for this protein in cell proliferation. Our data indicate that Fubp3 arginine dimethylation is significantly reduced in five-day differentiated cells (H/L = 18; Appendix A), suggesting that its (di)methylation (DiMA) status is critical for differentiation.Fubp3 possesses a KH domain, as well as the newly discovered SLiMs, EMIKKIQN, RCQHAARII, DYTKAWEEYY, and DYSAAWAEYY, whose role remains to be elucidated. Scrib is a protein similar to the Drosophila scribble protein, which functions as a scaffold in polarity, adhesion, synaptogenesis, and proliferation [83]. The mammalian Scrib protein, which is heavily dimethylated (DiMA) in differentiated cells (H/L = 0.009; see upregulated in DiMA, Appendix A), is involved in tumor suppression pathways, a function that may be mediated by its arginine methylation status. It possesses PDZ domains and LRR_8 repeats as well as the newly discovered SLiMs, WRCWR, CNFMQ, and GGWHN, whose functional role remains to be elucidated. Our observations implicate Fubp3 and Scrib arginine-dimethylation or monomethylation status in cell proliferation and differentiation and pave the way to the reconstruction of the networks that control these fundamental processes.

## 4. Materials and Methods

### 4.1. Reagents

Anti-methyl lysine or tyrosine antibodies (PTM-SCAN antibodies, pan-methyl lysine, mono-methyl arginine, symmetric di-methyl arginine, asymmetric di-methyl arginine, di-methyl lysine, and mono-methyl lysine) were purchased from Cell Signaling Technology (Danvers, MA, USA). The SILAC amino acids L-lysine (13C615N2, Cat# CNLM-291-632) and L-arginine (13C615N4, Cat# CNLM-539-H-1) were purchased from Cambridge Isotope Laboratories Inc., Cambridge, MA, USA. Trypsin treated with TPCK was obtained from Worthington Biochemical Co. (Lakewood, NJ, USA). Regular DMEM and DMEM deficient in L-lysine or L-arginine were obtained from Thermo Scientific. Fetal bovine serum (FBS), antibiotics, and phosphate-buffered saline (PBS, pH7.4) were obtained from Invitrogen (Carlsbad, CA, USA). Iodoacetamide and DL-dithiothreitol were purchased from Sigma–Aldrich (Burlignton, MA, USA).

### 4.2. Cell Treatments and Immunoblot Analysis of Whole Cell Extracts for Arginine- or Lysine-Methylated Proteins

The C2C12 myoblast cells were obtained from ATCC and maintained in high-glucose DMEM/20% FBS in 15 cm culture dishes (Corning, New York, NY, USA). The proliferating myoblasts were treated with 5 or 50 μM AdOx for 5 days, with fresh drug added every 2 days in fresh medium. Immunoblot analyses of the whole cell extracts isolated from the 5-day myoblasts or 5-day myocytes were carried out as previously described [84].

### 4.3. SILAC Labeling

Mouse muscle myoblast cells C2C12 were obtained from American Tissue and Cell Culture (ATCC, Manassas VA, USA) and maintained in DMEM supplemented with 20% FBS and penicillin/streptomycin at 37 °C and 5% CO2. For differentiation, the cells were grown to confluency in 15 cm dishes (Corning, NY), switched to DMEM with 2% FBS, and allowed to differentiate for five days. The proliferating C2C12 cells were labeled with SILAC amino acids. Briefly, the C2C12 cells were cultured in DMEM supplemented with 20% FBS and SILAC amino acids such as L-lysine (13C615N2) and L-arginine (13C615N4). After five doubling cycles, all the proliferating C2C12 cells were fully labeled. The SILAC-labelled proliferating C2C12 cells and the non-labelled differentiated C2C12 cells were prepared separately.

### 4.4. Peptide Sample Preparation for Whole Cell Proteomic Analysis

Three hundred micrograms of proteins (total 600 µg) from the proliferating C2C12 cells (SILAC-labeled) and the differentiated C2C12 cells (non-labeled) were mixed. The proteins were reduced with 5 mM dithiothreitol for 30 min at room temperature, alkylated with 10 mM iodoacetamide for 20 min in the dark, diluted with 20 mM HEPES pH8.0 to a final concentration of 1.5 M urea, and digested with sequencing grade trypsin (Promega, Madison, *WI, USA*) at a 1:30 *w*/*w* enzyme to protein ratio overnight at room temperature. The digested peptides were desalted using a Sep-Pak light cartridge and fractionated into 96 fractions by basic RPLC chromatography, as described previously [85,86]. These 96 fractions were concentrated into 12 fractions prior to drying under a vacuum. The samples were prepared in triplicate.

### 4.5. Peptide Sample Preparation for Anti-Arginine Immunopreciptation

The cells were washed with cold PBS three times, lysed in urea lysis buffer (20 mM HEPES pH 8.0, 9 M urea, 1 mM sodium orthovanadate, 2.5 mM sodium pyrophosphate, 1 mM ß-glycerophosphate), and sonicated twice on ice for 30 s, as described previously (Kim MCP 2014). The cell lysates were cleared by centrifugation at 12,000× *g* at 15 °C for 15 min. Equal amounts of protein from each proliferating and differentiated C2C12 cell line were mixed, reduced with 4.1 mM dithiothreitol for 30 min at room temperature, and alkylated with 8.3 mM iodoacetamide for 20 min in the dark. The proteins were diluted in 20 mM HEPES pH 8.0 to a final concentration of 1.5 M urea and immediately incubated with TPCK-treated trypsin at a 1:20 *w*/*w* enzyme to protein ratio at room temperature overnight with gentle end-to-end shaking. Theprotein digests were acidified by adding 20% trifluoroacetic acid (TFA) to a final concentration of 1% and subjected to centrifugation at 2000× *g* at room temperature for 15 min. The supernatant of the protein digests was loaded onto a Sep-Pak C18 cartridge (Waters, Cat# WAT051910, Milford MA, USA) equilibrated with 0.1% TFA for desalting. The peptides were eluted with 40% acetonitrile and 0.1% trifluoroacetic acid (TFA). The eluted peptides were lyophilized and subjected to immunoaffinity purification using anti-methyl-arginine or anti-lysine monoclonal antibodies. Three independent biological replicate experiments were performed and independent samples were prepared prior to peptide immunoaffinity purification (IAP).

### 4.6. Immunoaffinity Purification of Methylated Peptides

Immunoaffinity purification (IAP) of mono- or dimethylated arginine peptides was performed. After lyophilization, 30 mg of tryptic peptides was dissolved in 1.4 mL of IAP buffer (50 mM MOPS pH 7.2, 10 mM sodium phosphate, and 50 mM NaCl). After adjusting the pH to ~7.2 by adding 1 M Tris base, the peptides were centrifuged at 2000× *g* at room temperature for 5 min. Before IAP, the antibody-conjugated beads were washed twice with PBS and twice with IAP buffer at 4 °C. For IAP, the supernatant was incubated with beads conjugated with antibodies at 4 °C for 1 h, and the beads were washed twice with IAP buffer and thrice with cold ultra-high-purity water. The peptides captured by the antibody were eluted twice from the beads by incubating the beads with 60 µL 0.15% TFA at room temperature for 10 min. The eluted peptides were desalted using a STAGE tip, and the dried peptides were kept at −20 °C prior to the LC-MS/MS experiment.

### 4.7. Liquid Chromatography Tandem Mass Spectrometry

The LC-MS/MS analysis of three bRPLC replicates, each with 12 fractions, was carried out on an LTQ Orbitrap Elite mass spectrometer (Thermo Scientific, Bremen, Germany) interfaced with Easy nLC II (Thermo Scientific, Bremen, Germany). The peptides were loaded onto an enrichment column (75 µm × 2 cm), packed in-house (particle size 3 μm; ReproSil-Pur 100 Basic C18), and separated on an analytical column (75 µm × 10 cm; particle size 2.3 μm, eproSil-Pur 100 Basic C18) using a linear gradient of 7% to 45% solvent B (0.1% formic acid in 90% acetonitrile) for 100 min, with a total run time of 120 min. The mass spectrometer was operated in FT-FT mode. Survey scans and data-dependent MS/MS spectra were acquired at a resolving power of 120,000 and 30,000, respectively. Then, the 20 most intense precursor ions from a survey scan within an *m/z* range of 350 to 1800 were isolated with a 2 Da window and fragmented by HCD with 40% normalized collision energy. A dynamic exclusion of 45 s was used with a 7 ppm window. The internal calibration was performed using the lock mass option (*m*/*z* 445.1200025) in the ambient air.

### 4.8. Mass Spectrometry Data analysis

The Proteome Discoverer platform 2.0 suite (Thermo Fisher Scientific, Bremen, Germany) was used for the peak list generation and database searches. A precursor mass range of 350–8000 Da and a signal-to-noise ratio of 1.5 were the criteria used for the generation of the peak lists. The SEQUEST search algorithm was used for the database searches, with NCBI RefSeq 79 as the reference database (containing common contaminants). The oxidation of methionine and SILAC 2-plex (13C615N L-lysine and 13C615N4 L-arginine) was used as a dynamic modification. The carbamidomethylation of cysteine residues was used as a static modification. A maximum of one missed cleavage was allowed for tryptic peptides. Mass error windows of 5 ppm and 0.02 Da were allowed for the precursors and fragments, respectively. The decoy database was used to calculate the false discovery rate (FDR), with a cutoff value of 1%.

### 4.9. Identification of Enriched Domains and Short Linear Motifs (SLiMs) in MMA or DiMA Proteins

The top proteins in the MMA or DiMA samples that were down- or upregulated in the differentiated cells were analyzed for domain enrichment as follows:49 protein sequences (Appendix A) were downloaded from ENSEMBL and subjected to a domain search against the PFAM database using the pfam_scan software (http://ftp.ebi.ac.uk/pub/databases/Pfam/Tools/ (accessed on 2 August 2022)) (evaluation cut-off: 0.001). Next, the proteins of each of the four conditions (MMA up/downregulated and DiMA up/downregulated) were subjected to SLiM analysis. By using the MEME software [87], we searched for five motifs with a maximum width of 10 amino acids; the rest of the parameters were set to default. The position of each motif on the corresponding protein sequences was searched and visualized using the MAST software [88]. Both the MEME and the MAST algorithms are part of the MEME suite. The results of the MEME and MAST are in Appendix A, Figure 5A,B. The results of the PFAM search are in Appendix A.

### 4.10. Statistical Analysis

The statistical analysis of the experiments was performed with SigmaPlot 11 (Systat Software, created by *Leland Wilkinson, Chicago IL, USA*). *p* values of ≤0.05 were considered statistically significant. Three independent experiments were performed (n = 3), as indicated in Section 4.

## 5. Conclusions

In conclusion, our analysis of the global arginine methyloproteomics of the differentiation of mouse C2C12 myoblasts provides new insights into the heterogeneity of the activation status of methylases and demethylases. We identified several protein families whose arginine methylation status was correlated with proliferation or differentiation, which may be crucial for controlling the two opposing cellular phenotypes.The RRM proteins involved in RNA metabolism were prominent in all four classes of the modifications analyzed, in agreement with a previous report on RNA-binding proteins. The dimethylation status of the Fubp3 and Scrib proteins appears to correlate with differentiation because of their involvement in Myc networks (Fubp3) or in the formation of multiprotein scaffolding ensembles that control cell polarity and actin metabolism (Scrib). In general, our results on the differentiation of mouse C2C12 myoblast cells are in agreement with the previous studies that specifically implicated the RNA-binding proteins possessing RG/RGG-rich motifs. Our approach represents an effective method for identifying the novel arginine-methylated proteins which are critical for cell differentiation and possibly for cancer and for re-constructing the regulatory networks of proliferation and differentiation.

## Figures and Tables

**Figure 1 ijms-24-02124-f001:**
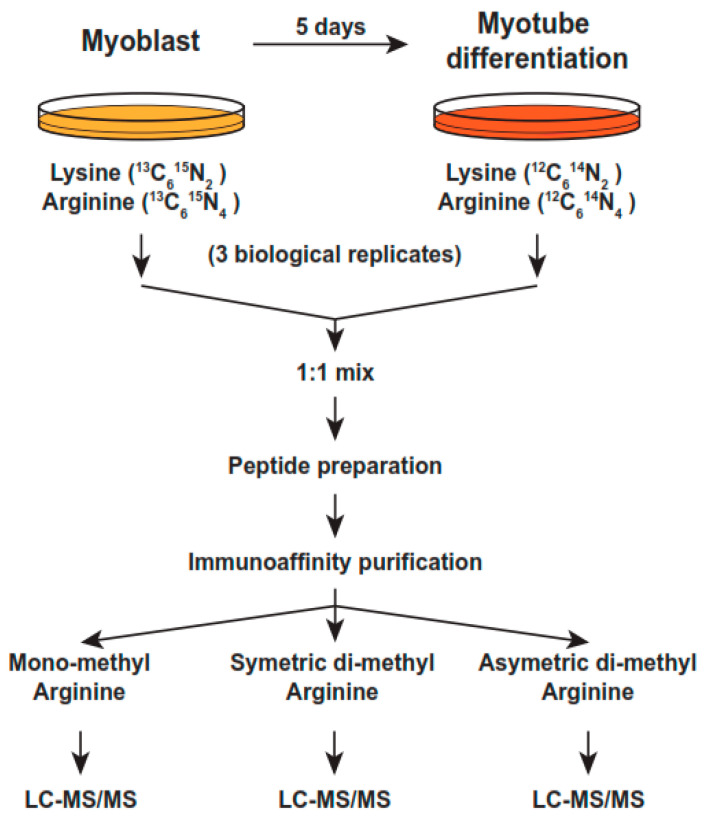
Strategy for SILAC methylproteomics in mouse C2C12 myoblast cells undergoing differentiation for 5 days. Proliferating C2C12 cells were expanded in glucose-rich medium with 20% FBS and in the presence of heavy, non-radioactive Lysine or Arginine. For differentiation, the cells were seeded in 15 cm dishes, in the same medium with 2% FBS. Triplicate experiments were performed and whole cell lysates (WCL) were pooled and protein concentrations were carefully determined with standard methods. For details, please see Section 4.

**Figure 2 ijms-24-02124-f002:**
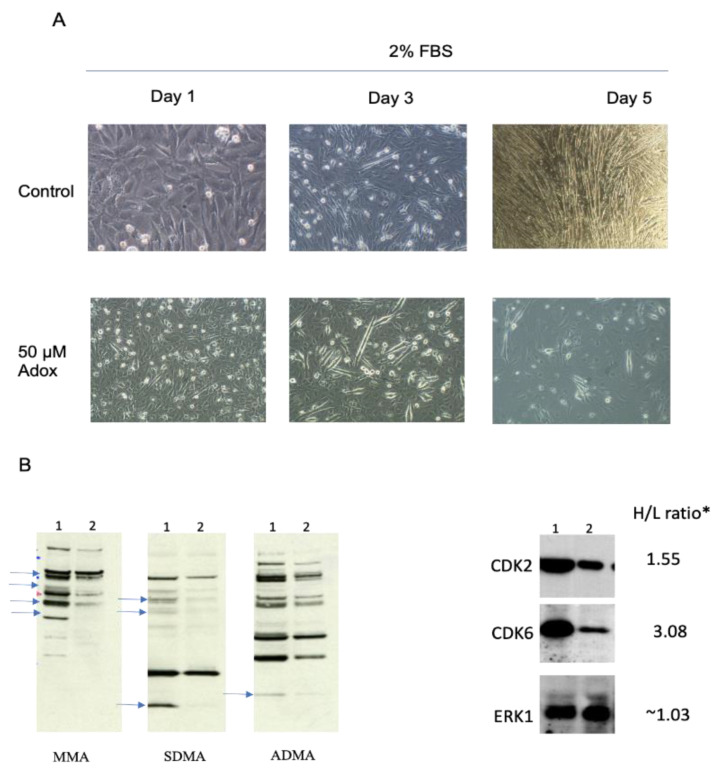
Morphological and immunoblot analysis of proliferating or 5-day differentiated C2C12 cells. Cells were plated at 40–70% confluence on 15cm dishes and proliferated in heavy, non-radioactive Lysine/Arginine medium for 5 generations to ensure full incorporation of these amino acids into proteins, as described in the Materials and Methods section. For differentiation, cells were plated at 30–40% confluence and placed in medium with 2% FBS. The cells were re-fed with fresh medium every 2 days, for detail see Materials and Methods. (**A**) Upper panels:control cells from days 1, 3, and 5 differentiating in light medium and 2% FBS. Lower panels:cells in 50 μM AdOX from days 1, 3 and 5. Magnification was at 40–100× and pictures were taken with a Nikon microscope, as described in the Materials and Methods section. (**B**) Left panel: immunoblots of monomethylated (MMA) and symmetric or asymmetric dimethylated protein arginine (SDMA and ADMA) in control (lane 1) or 5-day differentiated C2C12 cells (lane 2) with monoclonal anti-methylarginine antibodies. Right panels:immunoblots for CDK2, CDK6, or ERK1. Details in the Materials and Methods section. The H/L ratios are marked with an asterisk and indicated next to the proteins. Blue arrows point to the reduced arginine-methylated protein bands.

**Figure 3 ijms-24-02124-f003:**
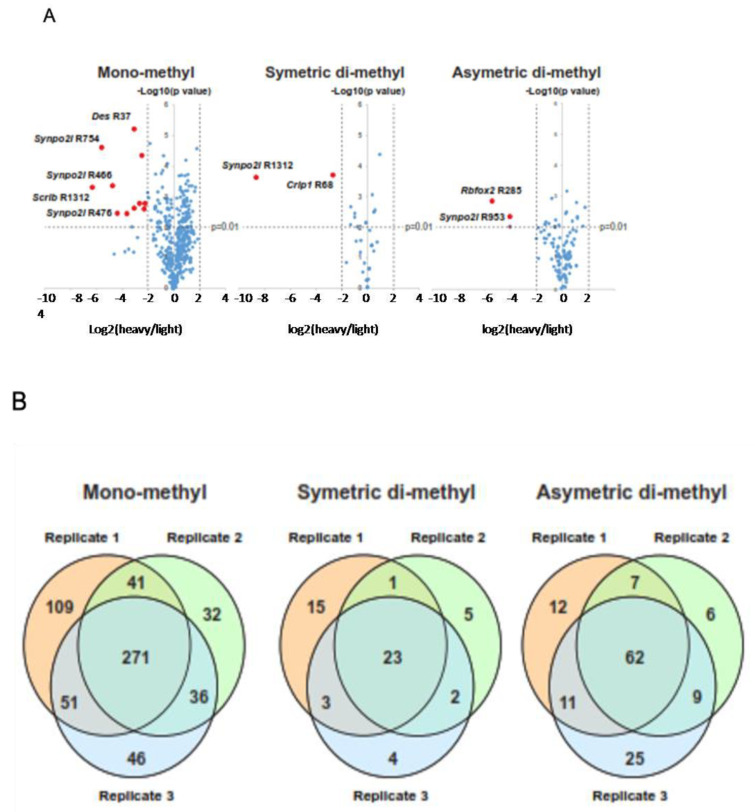
(**A**) Volcano plots of upregulated (red) or downregulated proteins (blue). Examples include DesR37 and Synpo2l, two muscle-specific proteins. Des, for desmin, is a gene encoding a muscle-specific class II intermediate filament which forms intracytoplasmic networks linking myofibrils to each other and to the cellular membrane. Familial cardiac and skeletal myopathy (CSM), as well as distal myopathies, are caused by mutations in this gene. Synpo2l is an actin-associated protein that modulates actin-based shape in sarcomeres; (**B**) Venn diagrams showing the overlapped identified proteins in three SILAC experiments (significance was set at a *p* value < 0.01).

**Figure 4 ijms-24-02124-f004:**
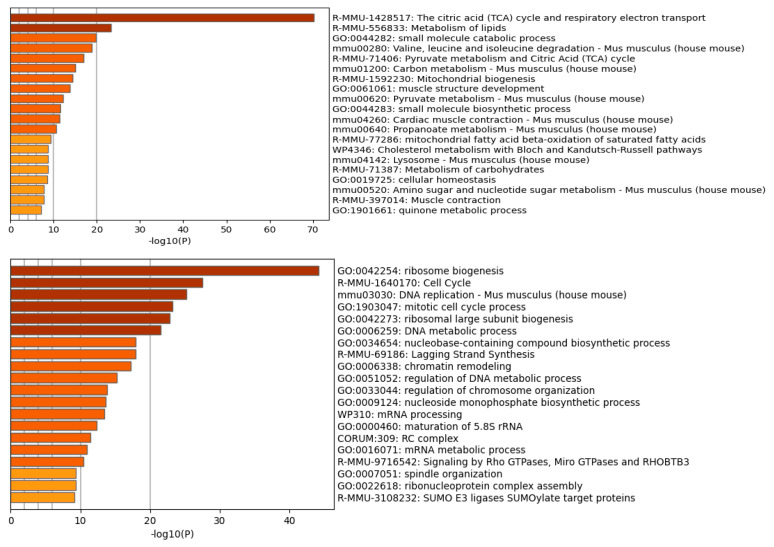
Enriched pathways were identified using Metascape [61]. The top upregulated pathways (**upper** panel) included respiration (citric acid cycle and respiratory electron transport), lipid metabolism, and pyruvate metabolism. Downregulated pathways (**lower** panel) in differentiated cells included ribosome biogenesis, cell cycle processes, and DNA replication.

**Figure 5 ijms-24-02124-f005:**
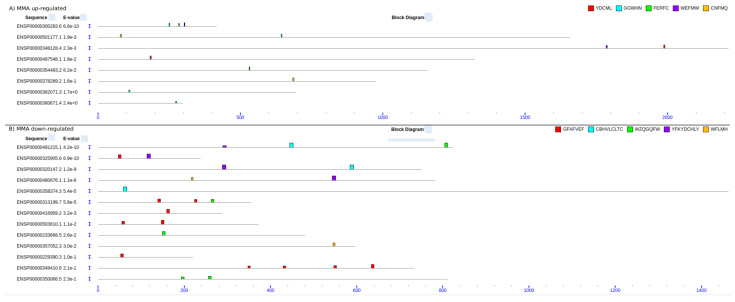
SLiMs in MMA of up- or downregulated proteins in differentiated cells. (**A**) SLiMs in MMA upregulated proteins. (**B**) SLiMs in MMA downregulated proteins. The motifs were searched and graphically visualized with the MEME and MAST software.

**Figure 6 ijms-24-02124-f006:**
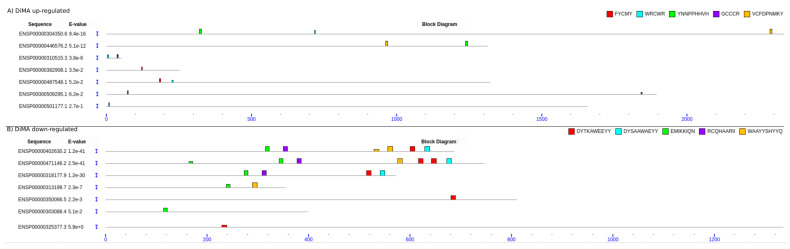
SLiMs in DiMA of up- or downregulated proteins in differentiated cells. (**A**) SLiMs in DiMA upregulated proteins. (**B**)SLiMs in DiMA downregulated proteins. The motifs were searched and graphically visualized with the MEME and MAST software.

**Table 1 ijms-24-02124-t001:** Top domains enriched in down- or upregulated MMA or DiMA samples in differentiated C2C12 cells.

MMA		Domain	Occurrences
	Downregulated	RRM_1	12
		WD-40	1
	Upregulated	PDZ	2
		MORN	1
		Mito-car	1
		RRM_1	1
DiMA			
	Downregulated	KH_1	12
		RRM_1	4
		DUF1897	4
		Mito-car	3
		WD-40	2
	Upregulated	RRM_1	10
		PDZ	4
		LRR_8	2
		MIF4G	2
		PAM2	2

## Data Availability

Data are available upon request.

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
