# Peer review of "The Dynamic and Crucial Role of the Arginine Methylproteome in Myoblast Cell Differentiation"

_ijms, 2023, doi:10.3390/ijms24032124_

Round 1

Reviewer 1 Report

The paper written by the following Authors: Nikolaos A. Papanikolaou, Marios Nikolaidis, Grigoris Amoutzias, Ariadni Fouza, Maria Papaioannou, Akhilesh Pandey and Athanasios G. Papavassiliou, entitled “The dynamic and crucial role of the arginine methylproteome in 2 myoblast cell differentiation” presents an interesting study of arginine methylproteome in 2 myoblast cell differentiation.

The paper is interesting, I believe that its present form may be presented in the journal.

Author Response

January 14, 2023

RE: REVISED RESEARCH Article (ijms-2137269 REVISED)

We would like to thank the Reviewer for his/her thoughtful evaluation of our manuscript and for his/her most welcome comments/suggestions. Accordingly, we have now revised our manuscript to reflect these comments.

In the revised Text all changes/additions/modifications made in response to the Reviewer’s points are marked up in red.

Please find below a point-by-point response to the issues raised by Reviewer 1:

Reviewer 1

The paper written by the following Authors: Nikolaos A. Papanikolaou, Marios Nikolaidis, Grigoris Amoutzias, Ariadni Fouza, Maria Papaioannou, Akhilesh Pandey and Athanasios G. Papavassiliou, entitled “The dynamic and crucial role of the arginine methylproteome in 2 myoblast cell differentiation” presents an interesting study of arginine methylproteome in 2 myoblast cell differentiation.

The paper is interesting, I believe that its present form may be presented in the journal.

Response: We thank the Reviewer for his/her positive evaluation of our work.

Trusting that we have addressed the Reviewer’s concerns, we would like to thank him/her for his/her help in improving our work.

With kind regards,

Prof. Athanasios G. Papavassiliou, MD, PhD

Prof. Nikolaos A. Papanikolaou, PhD

Corr

Reviewer 2 Report

The manuscript submitted by  Athanasios G. Papavassiliou and co-workers presents a very interesting set off analyses focused on defining the role arginine methylproteome in myoblast cell differentiation.

The presented study gives a valuable input into the molecular background of the myoblasts differentiation using C2C12 mouse cell model. The authors very well identify the need for identification of arginine methylation-driven effects in the myoblast differentiation process, as an attention of scientist have been mostly given to lysine methylation. The presented study base on the

 SILAC proteomics followed by high-resolution mass spectrometry, , which allows to suggest the potential targets for further analyses.

The introduction is very well written, coherent and very well brings to the topic of the paper. The methods for validation the hypothesis are properly chosen and described at the sufficient level of details. The analyses performed, are properly designed and controlled, replicated suitable, supported with well selected statistical tests. Presentation of data is clear. The conclusions reached are consistent with the presented data.

Comments;

-Some minor typos should be removed, e.g. in abstract (line 20) an extra space between C2C12 should be removed; line 197 – remove the dash inside the cell line; line 376 – add 2;

-It would be beneficial for the paper to create a scheme presenting a provisional mechanism of myoblast differentiation that includes arginine methylation.

Author Response

January 14, 2023

RE: REVISED RESEARCH Article (ijms-2137269 REVISED)

We would like to thank the Reviewer for his/her thoughtful evaluation of our manuscript and for his/her most welcome comments/suggestions. Accordingly, we have now revised our manuscript to reflect these comments.

In the revised Text all changes/additions/modifications made in response to the Reviewer’s points are marked up in red.

Please find below a point-by-point response to the issues raised by Reviewer 2:

Reviewer 2

The manuscript submitted by Athanasios G. Papavassiliou and co-workers presents a very interesting set off analyses focused on defining the role arginine methylproteome in myoblast cell differentiation.

The presented study gives a valuable input into the molecular background of the myoblasts differentiation using C2C12 mouse cell model. The authors very well identify the need for identification of arginine methylation-driven effects in the myoblast differentiation process, as an attention of scientist have been mostly given to lysine methylation. The presented study base on the SILAC proteomics followed by high-resolution mass spectrometry, which allows to suggest the potential targets for further analyses.

The introduction is very well written, coherent and very well brings to the topic of the paper. The methods for validation the hypothesis are properly chosen and described at the sufficient level of details. The analyses performed, are properly designed and controlled, replicated suitable, supported with well selected statistical tests. Presentation of data is clear. The conclusions reached are consistent with the presented data.

Response: We thank the Reviewer for his/her positive evaluation and kind words regarding our work.

Comments

-Some minor typos should be removed, e.g. in abstract (line 20) an extra space between C2C12 should be removed; line 197 – remove the dash inside the cell line; line 376 – add 2;

Response: We thank the reviewer for pointing out these typos. Accordingly, we have made the requested corrections.

-It would be beneficial for the paper to create a scheme presenting a provisional mechanism of myoblast differentiation that includes arginine methylation.

Response: We thank the reviewer for the suggestion. We had prepared a Graphical Abstract originally submitted with the manuscript (also uploaded with the revised manuscript). We feel that a specific mechanism is not warranted at present because the work focused on identifying potential arginine-methylated proteins that might be important in myoblast differentiation. This however remains to be pursued in future research.

Trusting that we have addressed the Reviewer’s concerns, we would like to thank him/her for his/her help in improving our work.

With kind regards,

Prof. Athanasios G. Papavassiliou, MD, PhD

Prof. Nikolaos A. Papanikolaou, PhD

Corresponding authors

Reviewer 3 Report

Review of the manuscript untitled : The dynamic and crucial role of the methylproteome in myoblast cell differentiation.

The manuscript addresses a global approach of proteomics to identify thechanges in the arginine methylproteome  during myoblast cell differenciation using the C2 C12 cell model.

The results are globally good and well presented, however some modifications are requested to improve the quality and clarity of the document.

Minor comments :

-          Do not put abbreviations in the abstract

-          Describe Slims meaning.

-          In the introductions PRMT family and type of methylation should be introduced and not in the discussion section.

-          Figure 1 is not well cited.

-          Please Remove figure 7.

-          Lane 93 : Set7 is not a histone methyltransferase but a lysine methyltransferase.

-          Lane 97 : is PRMT5 enzymatic activity required here ?

-          Lane 102 : it is not clear to speak about lysine methylation altought the authors investigate here arginine methylation.

-          Figure 2 : 50 mM ?

-          Lane 182 : gross ?

Major comments

-          Lane 164 : the authors can’t conclude that arginine methylation is required as Adox is a global inhibitor of all methylation reactions.

-          Some hits should be verified for the change of methylation level upon differentiation.

Author Response

January 14, 2023

RE: REVISED RESEARCH Article (ijms-2137269 REVISED)

We would like to thank the Reviewer for his/her thoughtful evaluation of our manuscript and for his/her most welcome comments/suggestions. Accordingly, we have now revised our manuscript to reflect these comments.

In the revised Text all changes/additions/modifications made in response to the Reviewer’s points are marked up in red.

Please find below a point-by-point response to the issues raised by Reviewer 3:

Reviewer 3

The manuscript addresses a global approach of proteomics to identify thechanges in the arginine methylproteome  during myoblast cell differenciation using the C2 C12 cell model.

The results are globally good and well presented, however some modifications are requested to improve the quality and clarity of the document.

Response: We thank the Reviewer for his/her positive evaluation of our work.

Minor comments

-Do not put abbreviations in the abstract.

Response: We thank the reviewer for pointing this out. Accordingly, we have defined abbreviations in the abstract.

-In the introductions PRMT family and type of methylation should be introduced and not in the discussion section.

Response: We thank the reviewer for the suggestion. We have removed the paragraph on PRMTs and transferred it to the Introduction section.

-Figure 1 is not well cited.

Response: We thank the reviewer for pointing this out. We have corrected this by eliminating “Figure 1” in line 43, after (PRMTs).

-Please Remove figure 7.

Response: We thank the reviewer for suggesting the removal of Figure 7. We have done so.

-Lane 93 : Set7 is not a histone methyltransferase but a lysine methyltransferase.

Response: We thank the reviewer for pointing this out. We have replaced “histone” with “lysine”.

-Lane 97 : is PRMT5 enzymatic activity required here ?

Response: We thank the reviewer for pointing this out. Indeed, as mentioned starting in line 104, PRMT5 is required for myoblast differentiation. We have included the appropriate reference (Ref. 53).

-Lane 102 : it is not clear to speak about lysine methylation altought the authors investigate here arginine methylation.

Response: We thank the reviewer for pointing this out. We have removed the reference on lysines from line 102 as instructed.

-Figure 2 : 50 mM ?

Response: We thank the reviewer for pointing this out. We have corrected to 50 μM.

-Lane 182 : gross ?

Response: We are grateful to the reviewer for pointing this out. We have corrected this by replacing “gross” with “genome-wide”.

Major comments

-Lane 164 : the authors can’t conclude that arginine methylation is required as Adox is a global inhibitor of all methylation reactions.

Response: We thank the reviewer for this observation. We have corrected the phrase by removing the reference to arginine methylation.

-Some hits should be verified for the change of methylation level upon differentiation.

Response: We thank the reviewer for the suggestion. We were unable to test other promising candidate proteins due to severe budgetary restrictions and therefore we had to restrict ourselves to testing cell-cycle kinases that are critical in cell differentiation.

Trusting that we have addressed the Reviewer’s concerns, we would like to thank him/her for his/her help in improving our work.

With kind regards,

Prof. Athanasios G. Papavassiliou, MD, PhD

Prof. Nikolaos A. Papanikolaou, PhD

Corresponding authors

Reviewer 4 Report

I checked your manuscript and described comments below.

I think that this paper does a very good job of analyzing protein arginine methylation using mouse C2C12 myoblast cells.

Certainly, the approach of this paper can identify arginine-methylated proteins that are related to cell differentiation and cancer, and I think it is an effective study of regulatory networks of proliferation and differentiation.

There is one problem. The supplementary file has not been uploaded, so I cannot confirm the complete contents.

I don't think this paper has any major mistakes or grammatical problems.

Author Response

January 14, 2023

RE: REVISED RESEARCH Article (ijms-2137269 REVISED)

We would like to thank the Reviewer for his/her thoughtful evaluation of our manuscript and for his/her most welcome comments/suggestions. Accordingly, we have now revised our manuscript to reflect these comments.

In the revised Text all changes/additions/modifications made in response to the Reviewer’s points are marked up in red.

Please find below a point-by-point response to the issues raised by Reviewer 4:

Reviewer 4

I checked your manuscript and described comments below.

I think that this paper does a very good job of analyzing protein arginine methylation using mouse C2C12 myoblast cells.

Certainly, the approach of this paper can identify arginine-methylated proteins that are related to cell differentiation and cancer, and I think it is an effective study of regulatory networks of proliferation and differentiation.

There is one problem. The supplementary file has not been uploaded, so I cannot confirm the complete contents.

I don't think this paper has any major mistakes or grammatical problems.

Response: We thank the Reviewer for his/her positive evaluation and kind words regarding our work. We have remedied this situation by uploading all five Supplementary files.

Trusting that we have addressed the Reviewer’s concerns, we would like to thank him/her for his/her help in improving our work.

With kind regards,

Prof. Athanasios G. Papavassiliou, MD, PhD

Prof. Nikolaos A. Papanikolaou, PhD

Corresponding authors

Reviewer 5 Report

The paper is of interest. Protein Arginine methylation is an extensive and functionally significant post-translational modification. However, little is known about its role in differentiation at the systems level. Using SILAC proteomics of whole proteome analysis in mouse C2 C12 proliferating or five-day differentiated myoblasts, followed by high-resolution mass spectrometry, biochemical assays, and specific immunoprecipitation of mono- or di-methylated arginine peptides, we identified several protein families that were differentially methylated on arginine. Our study is the first to reveal global changes in arginine mono- or dimethylation of proteins in proliferating myoblasts and differentiated myocytes and to identify enriched protein domains and novel SliMs. The data may be crucial for dissecting the links between differentiation and cancer growth.

The authors treated the cancer cells with Arg and Lys proteins, performed immunoblots SILAC labeling immunoprecipitation tandem mass spec and have applied statistical analyses. 

It contains 7 figures and 1 table and there are no major spelling/grammar errors thus it may be accepted for publication as it. 

Author Response

January 14, 2023

RE: REVISED RESEARCH Article (ijms-2137269 REVISED)

We would like to thank the Reviewer for his/her thoughtful evaluation of our manuscript and for his/her most welcome comments/suggestions. Accordingly, we have now revised our manuscript to reflect these comments.

In the revised Text all changes/additions/modifications made in response to the Reviewer’s points are marked up in red.

Please find below a point-by-point response to the issues raised by Reviewer 5:

Reviewer 5

The paper is of interest. Protein Arginine methylation is an extensive and functionally significant post-translational modification. However, little is known about its role in differentiation at the systems level. Using SILAC proteomics of whole proteome analysis in mouse C2 C12 proliferating or five-day differentiated myoblasts, followed by high-resolution mass spectrometry, biochemical assays, and specific immunoprecipitation of mono- or di-methylated arginine peptides, we identified several protein families that were differentially methylated on arginine. Our study is the first to reveal global changes in arginine mono- or dimethylation of proteins in proliferating myoblasts and differentiated myocytes and to identify enriched protein domains and novel SliMs. The data may be crucial for dissecting the links between differentiation and cancer growth.

The authors treated the cancer cells with Arg and Lys proteins, performed immunoblots SILAC labeling immunoprecipitation tandem mass spec and have applied statistical analyses.

It contains 7 figures and 1 table and there are no major spelling/grammar errors thus it may be accepted for publication as it.

Response: We thank the Reviewer for his/her positive evaluation of our work.

Trusting that we have addressed the Reviewer’s concerns, we would like to thank him/her for his/her help in improving our work.

With kind regards,

Prof. Athanasios G. Papavassiliou, MD, PhD

Prof. Nikolaos A. Papanikolaou, PhD

Corresponding authors
